# Effects of Low-Impact Development Facilities (Water Systems of the Park) on Stormwater Runoff in Shallow Mountainous Areas Based on Dual-Model (SWMM and MIKE21) Simulations

**DOI:** 10.3390/ijerph192114349

**Published:** 2022-11-02

**Authors:** Yue Lai, Yiyun Lu, Tingting Ding, Huiyi Sun, Xuanying Li, Xiaoyu Ge

**Affiliations:** 1School of Landscape Architecture, Beijing Forestry University, Beijing 100083, China; 2Chengdu Park Urban Construction and Development Research Institute, Chengdu 610000, China

**Keywords:** shallow mountainous areas, stormwater management, low-impact development, 1D SWMM model, 2D MIKE21 model

## Abstract

Rapid urbanization has triggered more serious urban flood risks. Many studies have focused on intra-urban flooding, but less attention has been paid to rainfall and flood risks at the urban fringe. Nowadays, China is vigorously promoting the construction of sponge cities in the whole area. It is important to study the construction of sponge cities in shallow mountainous areas, which are an important barrier between cities and mountains. The purpose of this paper is to investigate the performance of Low-Impact Development (LID) facilities under different rainfall scenarios in developed shallow mountainous areas. The second garden and flower exposition (“the Expo Park”) in Hebei Province is used as an example. The SWMM and MIKE21 models were used to simulate the hydrological processes before and after the construction of “the Expo Park”, and the models were calibrated with the measured data. Peak flow rate, outflow volume, rainfall-outflow ratio, runoff velocity, and water feature area of the water system were used as indicators to evaluate their effectiveness. The results showed that the placement of LID facilities had a positive impact on the construction of the shallow mountain area. Specifically, (1) LID facilities can reduce the peak flow rate, delayed peak flow time, outflow volume, and rainfall outflow ratio of stormwater runoff in mountainous areas; and (2) they can effectively collect rainwater and become a supplement to the landscape water system of the site. These findings provide a scientific basis for the construction of LID facilities in shallow mountainous areas, which is important for the development and flood management of shallow mountainous areas.

## 1. Introduction

Rapid urbanization and high-intensity human activities have significantly altered original natural hydrological cycle characteristics [1,2]. This has led to a series of problems such as increased surface runoff, frequent flooding and deterioration of the water environment [3,4]. In order to effectively control and solve the above urban problems, Chinese government ministry has proposed a new concept of urban stormwater management based on low-impact development, namely “sponge city”, in 2014 with reference to the experience and ideas of foreign stormwater management [5].

Currently, thanks to the pilot policy implemented in China, the construction of sponge cities in China has been effective on a small scale [6]. The sponge city construction areas are divided into key areas, which gives them the function of sponge cities as a priority [6]. However, from the perspective of the city as a whole, the sponge city construction areas are fragmented and relatively independent; they do not form an interconnected whole and lack a systemic nature [6]. In response to this phenomenon, the Ministry of Housing and Urban-Rural Development of the People’s Republic of China first proposed the “Notice on Systematic Territory-wide Demonstration of Sponge City Construction” in April 2021, and launched two batches of territory-wide sponge town construction demonstration cities in 2021 and 2022 [7,8,9]. The notice indicates the future direction of sponge city construction in China [9]. Territory-wide sponge city construction still follows the “Technical Guide for Sponge City Construction—Low Impact Development Rainwater System Construction (for Trial Implementation)” officially released by the Ministry of Housing and Construction in October 2014. Therefore, systematic region-wide construction of sponge cities based on low-impact development is imperative [9,10].

LID is a stormwater management approach that aims to protect the pre-development hydrology of a given site using decentralized micro-scale control measures [11]. There are various LID measures, including permeable paving, cisterns, green roofs, sunken green spaces, infiltration ponds, etc. [12,13]. Mainly through the processes of sedimentation, infiltration, filtration, retention and microbial degradation [13], the function of detaining and purifying rainwater is given free rein to achieve runoff reduction and reuse of rainwater, thus effectively reducing the risk posed by flooding [13]. LID systems are the core concept and means of stormwater management in sponge cities and will coordinate all aspects of urban development and construction [14]. Currently, many scholars have studied the application of LID measures in urban stormwater management [15,16]. Huang et al. [16] found that the combined deployment of LID facilities in communities had the best effect on runoff and pollutant abatement. Ahiablame et al. [17] used PCSWMM (PCSWMM, Guelph, ON, Canada) to simulate the effects of LID measures such as permeable paving and rain gardens on urban flooding, and the results showed that LID measures were effective in reducing runoff. Zhang et al. [18] found that the combined deployment of LID facilities can reduce the surface inundation area, reduce the depth of ponding, and delay the appearance of ponding in completed parks. In summary, LID facilities can effectively respond to urban flooding.

It also exposes the problems of LID facilities application scenarios in the process of sponge city construction at this stage. Most of the research on LID facilities and rainwater management is focused on cities, lacking exploration of LID facilities in urban fringe areas [6], and failing to effectively respond to the policy call of systematically promoting sponge city construction [9]. Due to the large span of the north-south dimension in China, the topography is complex and diverse, and the mountainous areas are vast. Many cities border mountain ranges, such as Qinhuangdao. These cities not only face drainage pressure from within the city but also suffer from rainfall and flooding pressure from the surrounding shallow mountainous areas, leading to more severe urban flooding [6]. As a transition zone between urban and mountainous areas, shallow mountainous areas serve as an ecological barrier for mountainous runoff to enter the urban and natural boundaries [19]. There is an urgent need to explore the role of LID facilities in coping with stormwater and flood risks in shallow mountainous areas.

Currently, domestic and international scholars have explored traditional flood control methods by comparing their flood control rates [20], flood modeling [21], and flood hazard prediction [22]. For example, Wyżga et al. [20] found that traditional valley flood control methods (levees) are not feasible in reducing stormwater costs and risks, and that flood control rates should be improved through floodplain storage and stormwater runoff storage. Tullos et al. [21] modeled floods through the HAZUS-MH model to assess disaster risk and flood response planning in watersheds and suggested adjusting land use regulations to reconnect floodplains and early warning systems. Guo et al. [22] explored rainfall distribution in mountainous regions of Siberia and its implications for hydrological hazard prediction in small mountainous watersheds. Research related to stormwater management in shallow mountainous areas has focused on underdeveloped mountainous areas, with studies focusing on stormwater hazard risk prediction [22], assessment systems, and management measures [21]. However, few studies have been conducted on the utility of LID facilities for stormwater and flood risk in developed shallow mountainous areas.

In this study, to fill the gap in the study of LID facilities in shallow mountainous areas, we propose a comparative research framework to study the performance of LID facilities in developed shallow mountainous areas under different rainfall scenarios, i.e., flood reduction effect and stormwater reuse utility, using “the Expo Park” located in developed shallow mountainous areas as an example. A one-dimensional hydrological model combined with a two-dimensional hydrodynamic model was used to simulate the hydrological processes before and after the construction of the shallow mountainous area. Five metrics were selected to evaluate the performance of the LID facility: peak flow rate, outflow volume, rainfall-outflow ratio, runoff velocity [23], and waterscape area of the water system. This is an important reference and guide for future flood management and construction in shallow mountainous areas.

## 2. Materials and Methods

### 2.1. Study Area

The city of Qinhuangdao is located in the eastern part of the famous Yanshan Mountains in China, and its low hills cover 72.8% of Qinhuangdao [24]. The study site is located at the junction of the Seaport and Beidaihe districts in Qinhuangdao City, Hebei Province, China, on the southeastern slope of Qiyun Mountain. The total area of the site is about 221.81 hm^2^, the area of “the Expo Park” is 120.17 hm^2^, and the catchment area of the surrounding areas such as Qiyun Mountain and urban roads is 101.64 hm^2^. The study area is shown in Figure 1. The topography of Qiyun Mountain is fragmented and the three existing gullies create significant runoff pressure from the mountainous area, and the Jinqin passenger railroad and residential areas in the east are under serious threat from stormwater flooding. The study area was mainly nurseries and fields before construction and was filled with potholes and ponds, woodlands, some farms and abandoned buildings.

Qinhuangdao has a temperate continental monsoon climate and is located in a semi-humid region of China [25]. According to multi-year data from meteorological stations, the average annual rainfall is about 645.9 mm. Seasonal rainfall is unevenly distributed, with 69.7% of annual rainfall concentrated in the summer, especially in July and August. The maximum summer precipitation can reach 1038.5 mm [26]. The high intensity of rainfall during short calendar periods leads to more severe flooding and debris flow problems in shallow mountainous areas. Therefore, it is necessary to study the utility of LID facilities for shallow mountainous areas and examine whether they can effectively regulate stormwater runoff in shallow mountainous areas and alleviate rainfall and flooding problems.

### 2.2. Data Collection

We collected the basic data for model construction, including historical Google satellite maps, soil data, relevant rainfall data, actual monitoring rainfall data, CAD topographic data, and information relating to the study area before and after construction. The historical Google satellite map was from Google Earth Pro software. The soil data were from the ground survey report of “the Expo Park”, and the rainfall data were from the National Meteorological Information Center, China Meteorological Data Network (http://data.cma.cn/ (accessed on 16 August 2022)). On 12 July 2021, the team members monitored the actual rainfall data in the study area. The topographic data (accuracy 1 m) and information about the study area before and after construction were obtained from the builder, Qinhuangdao City Landscape Bureau, and the topography after the design was obtained from the design team.

### 2.3. Research Methods

This study used a combination of one-dimensional and two-dimensional models to build hydrological and hydrodynamic models of the study area before and after construction. The accuracy of the models were then validated and calibrated with measured data to focus on flood risk and waterscape issues in natural shallow mountainous areas. The impact of LID facilities (park water systems) on stormwater runoff from hills was explored under the premise of meeting the Technical Guidelines for Sponge City Construction—Construction of Low-Impact Development Stormwater Systems (trial) (hereinafter referred to as the guidelines) [10]. The framework of the study is shown in Figure 2.

#### 2.3.1. Runoff Control Objective Identification

According to the Hebei Provincial Engineering Construction Standard “Technical Regulations for Sponge City Construction Projects” [DB13(J)/T210-2016], Qinhuangdao City adopted a newly revised formula for storm intensity [27]:q = 605.709 × (1 + 0.711 lgP)/(t + 1.040) × 0.464,(1)
where P represents the return period of the design rainfall (a), and t represents the confluence time (min).

According to the Technical Specification for Urban Flood Control (GB51222-2017) [18], the study area should be able to effectively respond to storms with return periods of 30 and 50 years to ensure that the site will not be significantly affected by rainfall events within the flood control criteria. Meanwhile, the design rainfall amounts under different return periods (namely 2, 5, 10, 30, 50, and 100 years) were determined by the Qinhuangdao City stormwater intensity equation based on this code, combined with the watershed area. The results were 104.33 mm, 128.64 mm, 147.03 mm, 176.19 mm, 189.74 mm and 208.13 mm, respectively. These were used as different scenarios for the model simulation.

#### 2.3.2. Calculation of Runoff Abatement from Internal and External Sources in the Area

According to the guidelines, when designing for the control of total runoff and runoff pollution, the design volume of storage should meet the index requirement of “control volume per unit area”, which is generally calculated using the volumetric method [10] with the following formula:V = 10 HφF,(2)
where V represents the design storage volume (in m^3^), H represents the design rainfall (in mm), φ represents the integrated rainfall runoff coefficient, and F represents the catchment area (in hm^2^).

According to the design rainfall standard of the study area H = 176.19 mm (30-year event) and the values of the rainfall–runoff coefficient of each catchment surface (Table 1), the weighted average calculation of the study area = 0.40, and the catchment area of the study area F = 221.81 hm^2^, from Equation (2), The total amount of runoff from internal and external sources of the site was calculated as 156,322.82 m^3^. According to the calculation results of the total amount of runoff and the overall layout of the water system, the maximum volume of designed landscape water bodies for storage was 171,976.85 m^3^, and the total area of landscape water bodies in the whole park was about 18.56 hm^2^.

#### 2.3.3. Rainfall Conditions of the Simulation Scheme

Regarding the rainfall conditions in the simulation scenario, a synthetic rainfall map was used as input conditions for the analysis. The rainfall map was derived from the analysis of rainfall numbers collected at the Qinhuangdao meteorological station. The synthetic rainfall map was calculated using the Chicago method based on the intensity-time-frequency relationship parameters for six regression periods (namely 2, 5, 10, 30, 50, and 100 years) [28,29]. Assuming a rainfall duration of 6 h and a temporal peak ratio of 0.4, Figure 3 shows the Chicago rainfall maps for the six selected return periods (namely 2, 5, 10, 30, 50, and 100 years).

#### 2.3.4. SWMM Model Construction

The EPA Stormwater Management Model (SWMM) is a dynamic rainfall–runoff simulation model used primarily for single-event or long-term (continuous) simulations of runoff flow and water quality in urban areas. It tracks the runoff flow and runoff water quality generated within each sub-catchment area, as well as the flow rate, water depth, and water quality in each pipe and channel over the simulation period, which consists of multiple time steps [30,31].

Pre-Construction Stormwater Modeling

On the basis of the pre-development topographic features of the study area, the site was simplified into 41 sub-catchment zones by following the principle of catchment zoning (Figure 4). According to the internal engineering geological survey report, the surface layer of the soil was vegetal fill, miscellaneous colored, loose, slightly wet, mainly composed of sandy soil, clay, and construction waste, with a thickness of about 0.6–0.7 m and a permeability coefficient of about 5.79 × 10^−4^ m/s. The second layer was powdery clay, brown, wet, containing sand and gravel, with a thickness of about 2.4–3.7 m and a permeability coefficient of about 1.16 × 10^−6^ m/s. The Horton infiltration model was selected on the basis of soil properties and model principles, with a maximum infiltration rate of 80–360 mm/h, a minimum infiltration rate of 5–50 mm/h, and an attenuation coefficient of 4 h^−1^. All other parameters were set according to the pre-development sub-bedding surface, with a comprehensive reference to the relevant literature and the recommended values in the SWMM user manual [30].

2.Post-Construction Stormwater Modeling

The developed model was designed vertically so that stormwater runoff from Qiyun Mountain, urban roads, and the site stagnated in the park’s landscape water system and was transferred through the central water vein to the end reservoir, which was docked to the surrounding stormwater municipal pipe network. The study area was divided into sub-catchment zones as a function of the current topographic slope, direction of catchment, and division principles, usually ranging in size from a few thousand square meters to several hectares [32]. The study area was simplified into 83 sub-catchment zones (Figure 5). According to the internal engineering geological survey report of the site, the surface layer of the soil was plantation soil, with a thickness of about 0.3 m. The second layer was powdery clay, brown, wet, containing sand and gravel, with a thickness of about 2.4–3.7 m and a permeability coefficient of about 1.16 × 10^−6^ m/s. On the basis of the soil characteristics and modeling principles, the Horton infiltration model was selected, and all other parameters were based on the sub-bedding surface, with a comprehensive reference to the relevant literature. The other parameters were set according to the conditions of the lower bedding surface, with reference to the relevant literature and the recommended values in the SWMM user manual [30].

The simulation results before and after the construction of the study area were compared and analyzed to explore the effects of LID facilities on peak flow rate and outflow volume.

#### 2.3.5. MIKE21 Modeling before and after Construction

MIKE21 is a distributed 2D hydrodynamic model that numerically solves the complete 2D Saint-Venant equations on a rectangular mesh [33]. At the same time, it numerically solves the Stokes equation with the assumption of hydrostatic pressure and the Boussinesq assumption [34]. The basic equations of the hydrodynamic module include the continuity and momentum equations. The two-dimensional MIKE21 model is mainly used to overcome the limitations of the one-dimensional SWMM model in modeling runoff velocity and inundation area. A comparative study of stormwater runoff velocity before and after the construction of the study area allowed for the investigation of the reduction in maximum velocity by LID facilities and the classification of barge types and structures according to the velocity. Simulation of the inundation area allowed investigating whether the landscape system in the park could be replenished with stormwater alone to meet the water demand and drainage time of the park during the Expo.

In this study, the MIKE 21 model and the SWMM model maintained the consistency of rainfall events, simulation times and time steps.

(a)MIKE21 Pre-Construction Stormwater Modeling

The simulation area was determined according to the catchment area, the model grid was established in conjunction with the pre-construction site in the vertical dimension. The parameters that need to be input to the pre-construction MIKE 21 model include study area mesh file, simulation time, solution technique, flood and dry, precipitation-evaporation, and infiltration. Topographic interpolation was performed to mesh the triangular grid of the study area, and the maximum element of the grid was set to 150 m. In flood and dry, the drying depth is 0.0001 m, and the wetting depth is 0.002 m. The rainfall input is the rainfall for the return period of six rainfall events calculated by the Chicago algorithm. Evapotranspiration data are derived from the Qinhuangdao weather station. The infiltration data are derived from the simulated infiltration from the pre-construction SWMM model. Other general parameters were referred to in the MIKE 21 User Manual [33]. Figure 6 shows the pre-construction MIKE 21 model software interface for the study area.

(b)MIKE21 Post-Construction Stormwater Modeling

The post-construction MIKE 21 model requires the same parameters to be entered as the pre-construction model parameters. The mesh file production, rainfall-evaporation data, and infiltration data differ from the pre-construction model. Mesh files are produced by entering post-construction topographic data (1 m accuracy) and the maximum element of the grid is set to 150 m. Rainfall event simulations add monitoring data from 1 June to 31 October 2018. Infiltration data were derived from the calibrated post-construction SWMM model simulations of infiltration. Figure 7 shows the post-construction MIKE21 model software interface for the study area.

Both pre-and post-construction models are to simulate the rainfall event return periods (namely, 2, 5, 10, 30, 50, and 100 years) for the study area to investigate the role of LID facilities on stormwater runoff flow rates. The simultaneous post-construction model combines the season of “the Expo Park” (18 July to 31 October) [35] and the rainfall characteristics of the semi-humid region to simulate the area of the water system in the study area after construction using rainfall and evaporation data from 1 June to 31 October 2018. The inundated area of the water feature on the site was counted at 8:00 p.m. each day to map the process of water feature inundation changes during the opening period and to investigate whether the park’s collected stormwater runoff could meet the water landscape needs.

#### 2.3.6. Calibration of the Constructed SWMM Model and MIKE21 Model with Measured Data

We only calibrated the post-construction SWMM model due to the unavailability of measured data prior to the construction of the study area. To calibrate the predictive capability of the post-construction stormwater management model for the study area, rainfall data for 12 July 2021 were selected for model calibration. The rainfall at this location lasted for 24 h and reached 92.52 mm. The actual measurements at the site lasted 2.5 h with a 15-min interval. After several trials and errors [36], a set of parameters reflecting the process production at the site was obtained, as shown in Table 2.

In this study, three indices were selected to assess the simulation results of model calibration and validation. These were the coefficient of determination (*R*^2^), the root-mean-square error (*RMSE*), and the Nash–Sutcliffe efficiency coefficient (*NSE*). *R*^2^ shows the degree of fit of the measured and predicted data. *RMSE* indicates the difference between measured and predicted values. *NSE* is a reliable criterion for assessing the predictive capability of a hydrological model. If the values of *R*^2^, *RMSE*, and *NSE* are close to 1, 0, and 1, respectively, the prediction model performs best [37]. Moreover, according to the current criteria for assessing the effects of sponge cities, the *NES* value for model calibration and validation should not be 0.5 [38]. These criteria are defined as follows:(3)$$R^{2}=\frac{\left(\sum_{i=1}^{n}(y_{i}-\bar{y}\right)(y_{i}^{\prime }-\bar{y\prime }){)}^{2}}{\sum_{i=1}^{n}{\left(y_{i}-\bar{y}\right)}^{2}\sum_{i=1}^{n}{\left(y_{i}^{\prime }-y\prime \right)}^{2}},$$
(4)$$RMSE=\sqrt{\frac{\sum_{i=1}^{n}{\left(y_{i}-y_{i}^{\prime }\right)}^{2}}{n}},$$
(5)$$NSE=1-\frac{\sum_{i=1}^{n}{\left(y_{i}-y_{i}^{\prime }\right)}^{2}}{\sum_{i=1}^{n}{\left(y_{i}-\bar{y}\right)}^{2}},$$
where $y_{i}$ is the monitored data, $\bar{y}$ is the average of the monitored data, $y_{i}^{\prime }$ is the predicted data, $\bar{y^{\prime }}$ is the average of the predicted data, and *n* is the number of data.

Figure 8 shows the monitoring and simulation hydrograph results of the SWMM model calibration using the measured data on 12 July 2021. The three indices *R*^2^, *RMSE*, and *NSE* for evaluating the simulation results of SWMM model calibration and verification were 0.96, 0.11, and 0.92, respectively. Although the values of *R*^2^, *RMSE*, and *NSE* were infinitely close to 1, 0, and 1, respectively, a slight difference between the monitored value and the simulated value can be seen in Figure 8. This difference was mainly due to the short preheating time of the SWMM model and the unstable monitoring data. Generally speaking, the results of SWMM model calibration and verification were satisfactory, and the established model could be used in this paper.

As there are no detailed measured data to support the verification of the MIKE 21 model, the preliminary accuracy of the MIKE 21 model was judged by the simulation results of the calibrated SWMM model. Initial calibration was used to verify the accuracy of the MIKE 21 model by simulating the storm event on 12 July 2021 and comparing the results of the yield flow simulated using the MIKE 21 model with the yield flow simulated using the SWMM model. The results showed that the runoff produced by the SWMM model was 108,691.34 m^3^, and the runoff produced by the MIKE 21 model was 110,784.00 m^3^. The difference in runoff results was small, mainly caused by the different operation mechanisms of the two models. Generally speaking, the established MIKE21 model could be used in the research of this paper.

## 3. Results

### 3.1. Peak Flow Rate

Under different return periods of rainfall events, LID facilities have good regulation effects on heavy rainfall and flooding. It can effectively reduce the flood flow, delay the flood time, protect shallow mountainous areas and surrounding cities from rainfall and flooding, and relieve urban flood pressure. Table 3 shows the peak flows, peak occurrence times and peak flow reduction rates before and after construction in the study area for selected rainfall event return periods (namely 2, 5, 10, 30, 50 and 100 years). At two years’ return period, the peak flow is zero and the peak flow reduction rate reaches 100% after construction, i.e., the flood peak disappears. The results are similar to the findings of Yin [39] et al. where there was essentially no significant runoff in the study area under small rainfall events. Flood flows were reduced by 64.50% to 100.00% under all rainfall event return periods. This is consistent with the findings of Lucas [40] and Guo [41] et al. who found that peak flows were reduced by 53~87.5% through the implementation of LID facilities. In addition, the runoff control capacity of LID facilities decreased as the intensity of rainfall increased. This is consistent with previous studies [42,43]. Compared to the pre-development results, the peak flows all occurred with a delay ranging from several hours. This differs significantly from the results of studies within cities. Yang [44] et al. found that the time of peak flow emergence remained constant after the sponge city was built. This shows that the deployment of LID facilities in shallow mountainous areas will be better in terms of peak emergence time compared to intra-city. The reason for this result may be that the deterrent effect of LID facility facilities on stormwater may be offset by the accelerating effect of the inner city drainage network [44]. Figure 9 shows the simulated hydrological processes for the return period of six rainfall events in the study area.

### 3.2. Outflow Volume

As shown in Table 4, reduction rates decreased by 33.88% to 100% under different rainfall event return period conditions. The longer the rainfall return period, the lower the reduction rates and the smaller the decrease. This finding supports the results of Guo [41] et al. regarding the extent of outflow volume reduction for LID facility practices. It is noteworthy that reduction rates suddenly decrease when the rainfall return period reaches 30 years. Therefore, it is reasonable to conclude that the design criteria for flood control in shallow mountainous areas should also include return periods greater than 30 years, which is consistent with the existing criteria for central urban areas in Hong Kong, the United States, and the United Kingdom [45,46].

### 3.3. Rainfall–Outflow Ratio

We introduced the rainfall–outflow (RO) ratio to assess the ability of catchments to absorb rainwater. RO represents the percentage of total rainfall in the catchment. Simply put, it means how much rain is extracted or retained in the catchment area for each storm event. A lower RO ratio indicates that more rainfall is stored in LID facilities built in the catchment. For higher RO ratios, the situation is just the opposite, indicating that catchments are not very effective at controlling current speed, which may lead to higher flood risk in and downstream of the catchment [47]. The calculation formula is as follows [47]:(6)$$RO=\frac{\sum C_{outlet}}{C_{rainfall}},$$
where $\sum C_{outlet}$ is the sum of the water volume at the outlet of the catchment (m^3^), and $C_{rainfall}$ is the total rainfall in the catchment area during the rainstorm (m^3^).

The RO ratios of the six storm types before and after the construction of the study area are drawn in Figure 10. As can be seen from the figure, the RO ratio after construction decreased compared with that before construction, and the RO ratio under six storm events decreased by 10–15%. This shows that the setting of LID facilities effectively reduced the relative proportion of rainfall reaching the drainage outlet of the catchment area, with a higher reduction rate for the drainage outlet with frequent rainfall. This result is similar to the discussion of Alexander [47] et al. on the utility of water storage in green spaces in catchments.

### 3.4. Runoff Velocity

Under the same rainfall condition in the return period, the change in runoff velocity before and after construction was simulated and compared using MIKE21 software. The comparison of runoff velocity before and after construction was divided into two aspects: (1) comparison of the maximum velocity before and after the construction (Figure 11), where it can be seen from the figure that the runoff velocity in the site after the construction was less than 3 m/s, with a larger runoff velocity (0.5–3 m/s) distribution after the construction in line with the water system distribution in the site; (2) comparison of runoff velocity reduction before and after construction (Figure 12). The runoff velocity of each grid with the same step size in the research area before and after construction was compared to obtain the runoff velocity difference of each grid before and after construction in the 720-step research area, and then the maximum runoff velocity difference of each grid was screened out. As shown in Figure 11, compared with the site runoff velocity before construction, the site runoff velocity after construction was reduced by about 0–1.5 m/s. The above two points show that LID facilities (park water system) could effectively reduce and control the flow rate.

### 3.5. Waterscape Area of Water System

“The Expo Park” was held from 16 July to 18 October 2018 [35]. This paper simulated the waterscape area of the site from 1 June to 31 October 2018, and the total rainfall in the simulated period was 621.4 mm. According to the analysis of the results obtained for “the Expo Park” (95 days in total), the submerged area of water could reach 60% of the total waterscape area (18.56 hm^2^), with 80% of the total waterscape area covered in 72 days (Figure 13). The submerged area distribution of the landscape water system in the study area is shown in Figure 14 The rainwater runoff collected by the internal water system of the park could basically ensure that the whole landscape water system was kept in a relatively stable state during “the Expo Park”, thus meeting the demands of the water landscape.

## 4. Discussion

First, extreme precipitation events are a cause of flood risk in shallow mountainous areas, mainly due to the convergence of high-intensity rainfall into an explosive surface runoff in a very short period of time [48]. Simulation of surface runoff movement processes is essential. However, the simulation of runoff velocity proposed in this study cannot be satisfied by the SWMM hydrological model alone. Therefore, to overcome the shortcomings of SWMM in simulating hydrodynamic processes, this study combines the hydrodynamic module of MIKE21 to simulate site runoff velocities [49]. At the same time, MIKE21 can simulate the extent of stormwater inundation in the site [33], which helps to study another index: the Waterscape Area of the Water System, i.e., the effect of stormwater reuse.

Second, with the rapid urbanization in China, controlling surface runoff is an important goal for urban stormwater management [50,51,52]. LID facilities are an important measure to control surface runoff to mitigate flooding [53]. Many previous studies have argued that LID facilities can significantly reduce surface runoff. Huang et al. [16] showed that LID facilities can reduce the amount of runoff in a community by 64.77% to 58.27%. Zhang et al. [18] found that LID facilities delayed the appearance of ponding at waterlogged points in the park by 10–30 min. consistent with the results of this study, LID facilities reduced peak flood flows in developed shallow mountainous areas by 100.00%−64.50% and delayed the appearance of peak flows by 45–125 min. Effective reduction of peak flows and delayed appearance of peak floods through LID measures can reduce the risk of flooding caused by high-intensity rainfall converging into explosive surface runoff within extreme events [48]. The difference is that under low-intensity rainfall conditions, the deployment of LID facilities in shallow mountainous areas can make the flood peaks disappear completely and the performance of LID is stronger than that of urban areas. Since the LID facility in this study is the water system of the “Garden Expo”, the scale of the LID facility is much larger than that of the LID facility in the urban area, and the amount of rainwater that can be stored and detained is greater than that under low-intensity rainfall conditions. This is consistent with previous studies that have found that increasing the size of LID facilities can reduce peak runoff flows [54]. Wang [55] and Zhao [56] et al. found that the maximum flow rates in most urban areas were small. The possibility of flow velocity loss before and after construction is small. This is different from the results of this study. The runoff velocities of the sites after construction in developed shallow mountainous areas were all reduced by 0–1.5 m/s. The LID facilities had a significant mitigating effect on the flow velocities of surface runoff in shallow mountainous areas. Explosive surface runoff is more likely to create flood risk [48]. Therefore, controlling the flow rate of surface runoff can reduce the flood risk. We failed to further investigate the causes of the difference in runoff velocity between shallow mountainous areas and urban areas, and future studies can explore the factors affecting the performance of LID facilities in shallow mountainous areas and urban areas, and investigate the methods to optimize the performance of LID facilities in different areas.

Due to the study’s limitations, pre-construction field flow measurements were unavailable for this study, and the SWMM and MIKE21 models could not be validated and checked before construction. Therefore, the accuracy of the pre-construction simulation results was not verified. We calibrated the models after construction, and although the simulated runoff curves closely matched the observed data, the limited number of monitoring events may have reduced the calibrated parameters’ accuracy under various storm conditions. It is regrettable that due to various external conditions, such as COVID-19, we were unable to obtain more monitoring data under different rainfall events and validate the calibrated model and improve the accuracy of the model. There is a possibility that the results of the model simulation and the actual results have large errors, resulting in the failure of this study. However, the research framework of this study on the performance of LID facilities in shallow mountainous areas is feasible. This is still an important reference and guidance for future flood management and construction in shallow mountainous areas. Meanwhile, the mesh size setting of the MIKE21 model could also affect the accuracy of model simulation. Determining the appropriate grid size in a two-dimensional flood simulation is the key to optimizing the performance of the model, so as to obtain accurate results in the shortest time [39]. In this paper, the grid setting adopted the empirical value corresponding to the study area, and a variance comparison of different grid sizes and their model accuracy was not performed.

In future studies, we will seek to improve the completeness of monitoring data under different rainfall events and to improve the accuracy of the model. The performance of LID facilities in shallow mountain areas will be further compared in different climatic zones.

## 5. Conclusions

In this study, we combined the SWMM and MIKE 21 model to simulate the surface flow and inundation process before and after the construction of shallow mountain areas in different repetition periods. The main conclusions are as follows:(1)In terms of flood risk, LID facilities can effectively reduce the peak flow rate of mountain rainwater runoff and delay the time of peak flow, as well as the outflow volume and the rainfall–outflow ratio, thus enhancing the ability of the catchment area to absorb and maintain rainwater. This can effectively reduce the maximum current speed in most sites. Further analysis shows that the design standard for waterlogging control in shallow mountain areas should also include a return period of more than 30 years.(2)In the area of recycling rainwater for water landscape creation, the setting of LID facilities can effectively collect rainwater, become a supplement to the landscape water system in the site, and meet the water landscape requirements of “the Expo Park”. Furthermore, the simulation of the submerged process can effectively predict the submerged range of the waterscape in the design site under different rainfall conditions, which has certain reference value for waterscape design.

These findings provide a scientific basis for the construction of LID facilities in shallow mountainous areas, which is important for the development and flood management in these areas.

## Figures and Tables

**Figure 1 ijerph-19-14349-f001:**
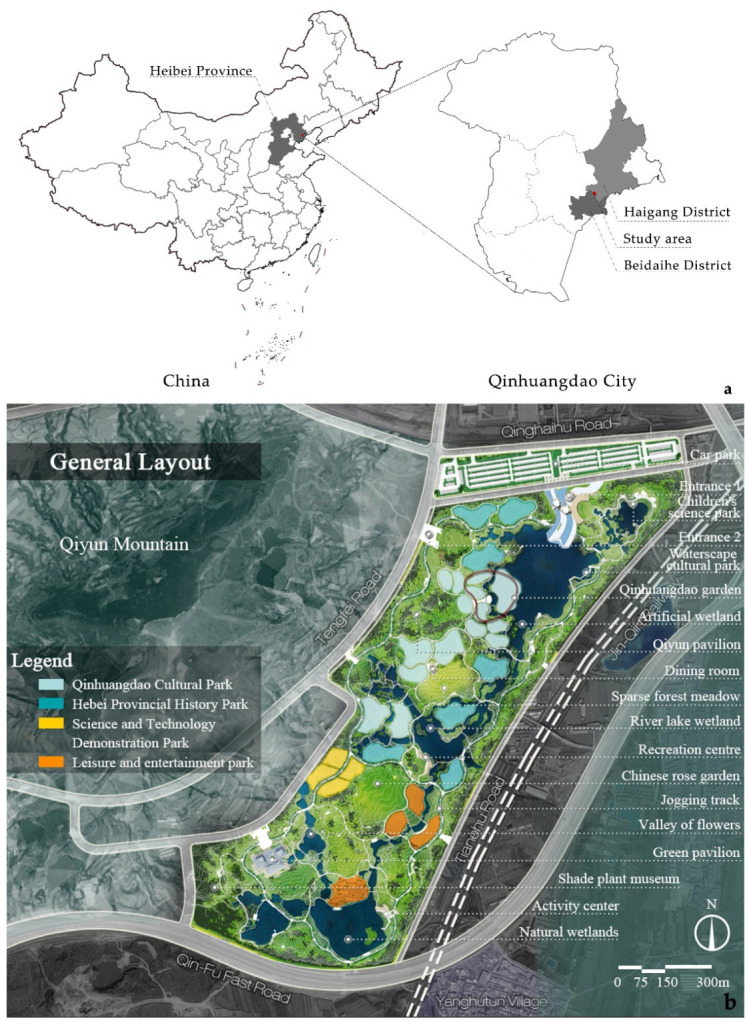
(**a**) Geographical location of the study area. (**b**) Study Area Plan.

**Figure 2 ijerph-19-14349-f002:**
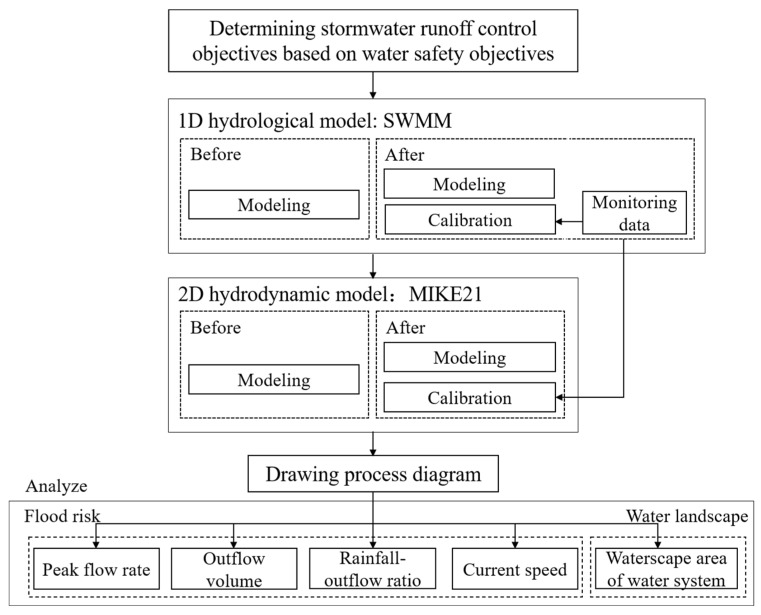
Research framework.

**Figure 3 ijerph-19-14349-f003:**
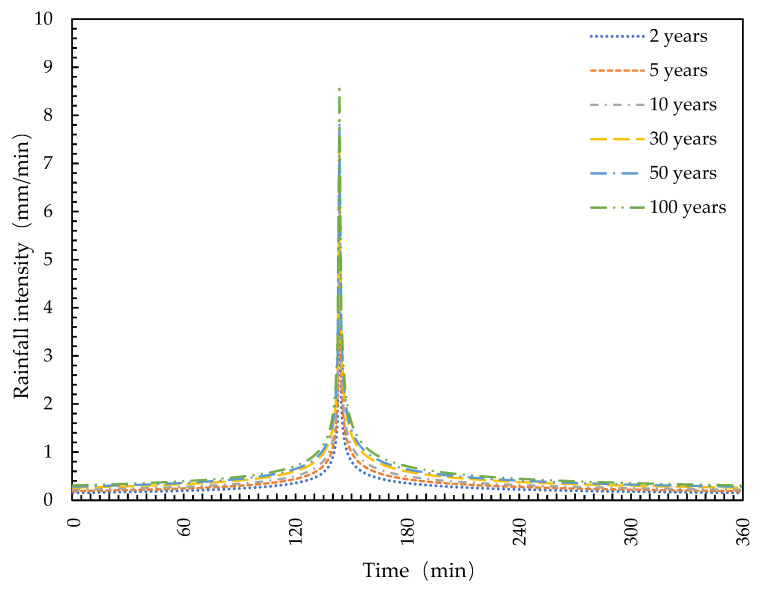
Design rainfall under different return periods of rainfall events in Qinhuangdao City.

**Figure 4 ijerph-19-14349-f004:**
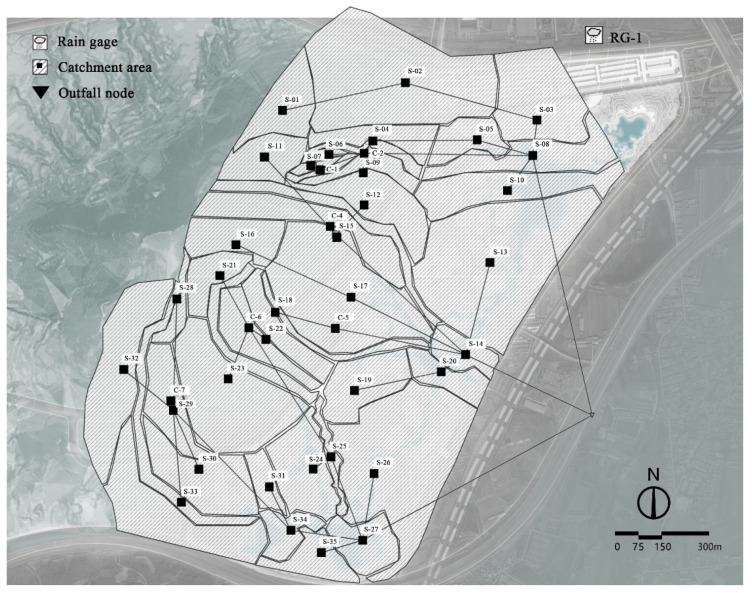
Subdivision map of site catchment area before construction.

**Figure 5 ijerph-19-14349-f005:**
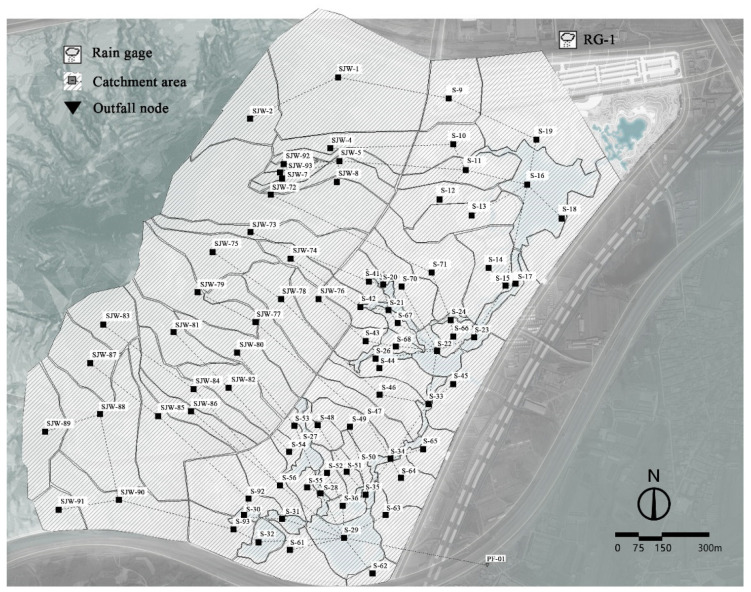
Subdivision map of site catchment area after construction.

**Figure 6 ijerph-19-14349-f006:**
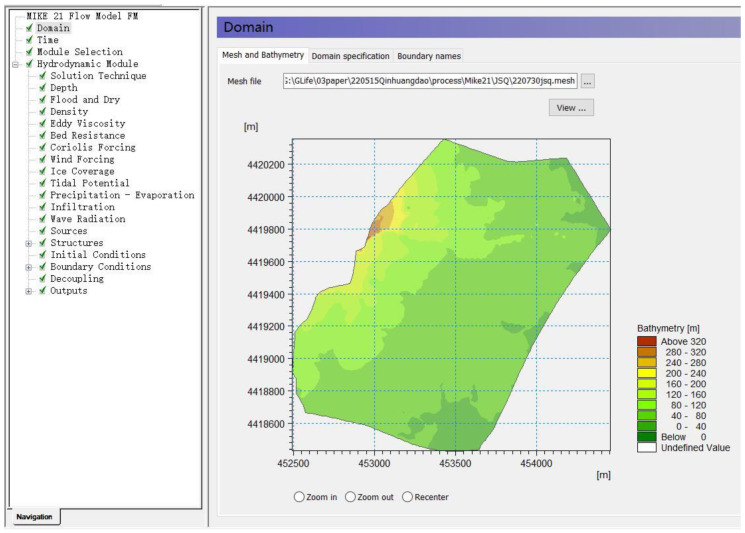
MIKE 21 model built before site construction. Notes: MIKE 21 Flow Model FM is a modelling system based on a flexible mesh approach.

**Figure 7 ijerph-19-14349-f007:**
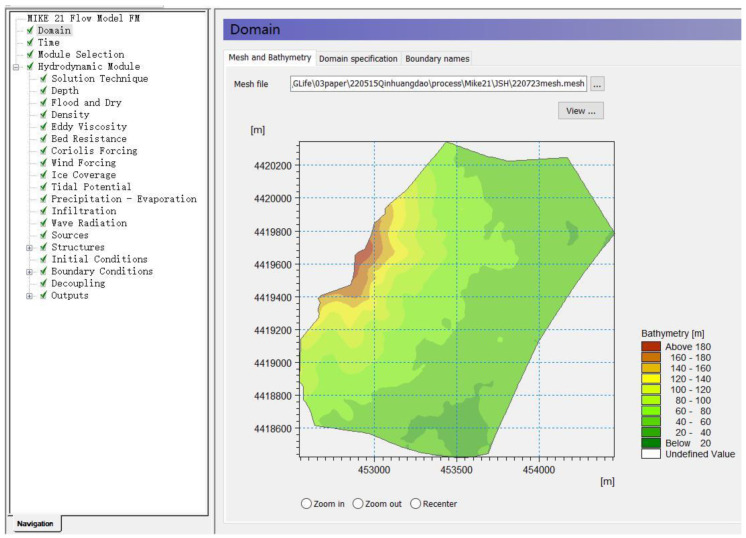
MIKE 21 model built after site construction. Note: when the MIKE 21 model simulates flow velocity and submerged area, its rainfall and evaporation parameters are set differently. Notes: MIKE 21 Flow Model FM is a modelling system based on a flexible mesh approach.

**Figure 8 ijerph-19-14349-f008:**
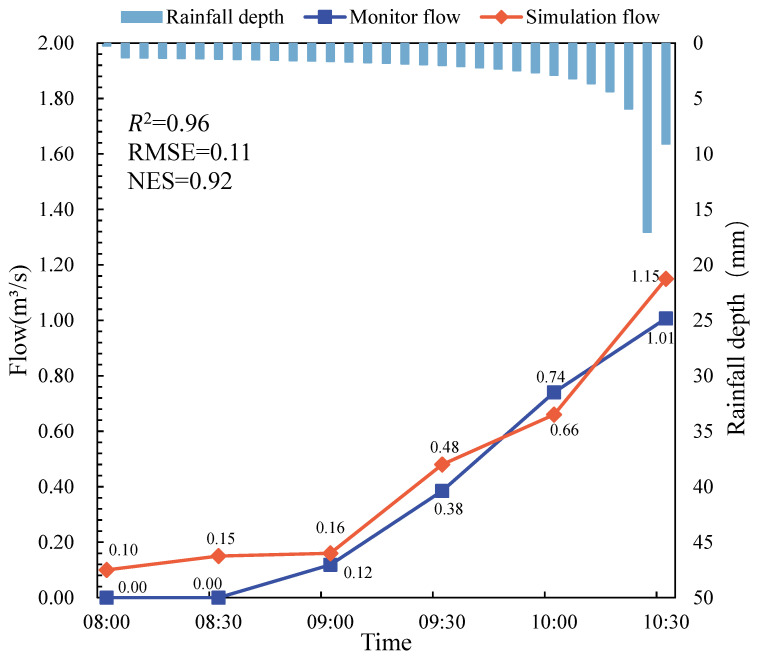
Process line of simulated and observed values on 12 July 2021.

**Figure 9 ijerph-19-14349-f009:**
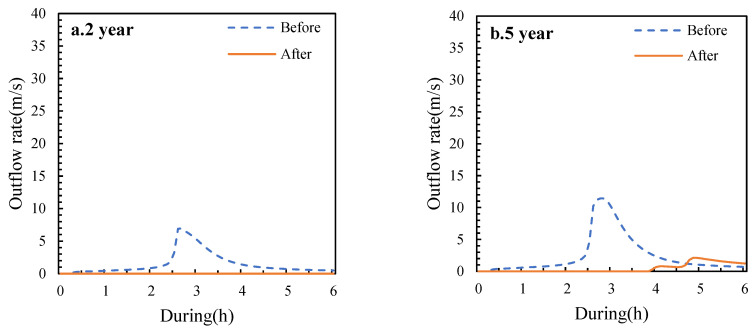
The outflow process of rainfall in different return periods before and after the construction of the research area.

**Figure 10 ijerph-19-14349-f010:**
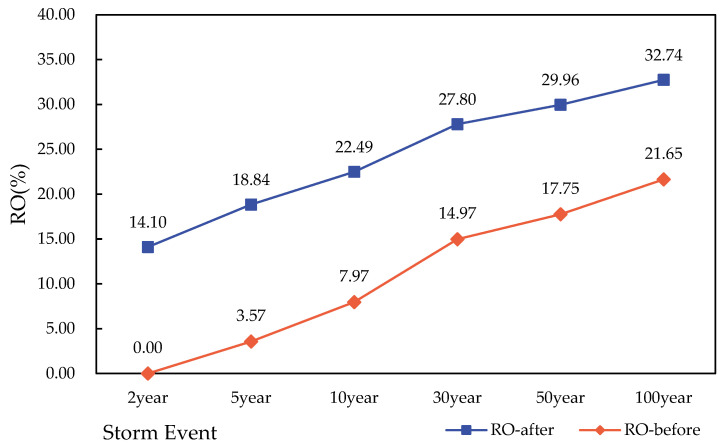
Rainfall–outflow ratio before and after the construction of the study area under different rainstorm events.

**Figure 11 ijerph-19-14349-f011:**
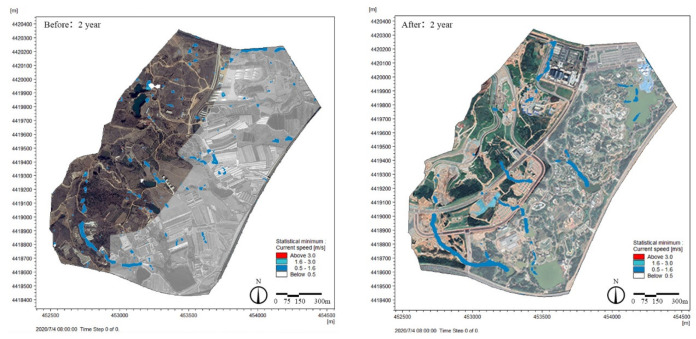
Comparison of maximum flow velocity under rainfall in different return periods before and after construction. Notes: The comparison chart of other rainfall events is annexed.

**Figure 12 ijerph-19-14349-f012:**
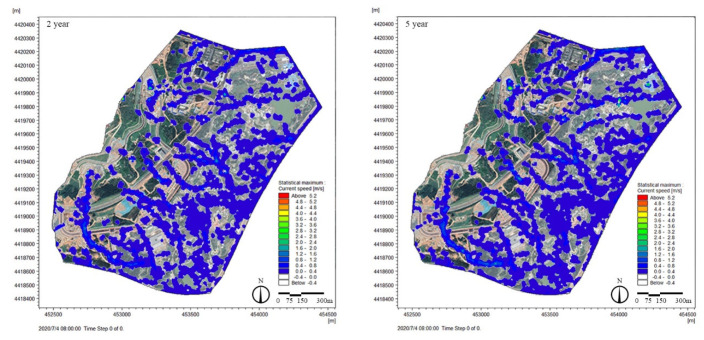
Changes in current speed under rainfall in different return periods before and after construction.

**Figure 13 ijerph-19-14349-f013:**
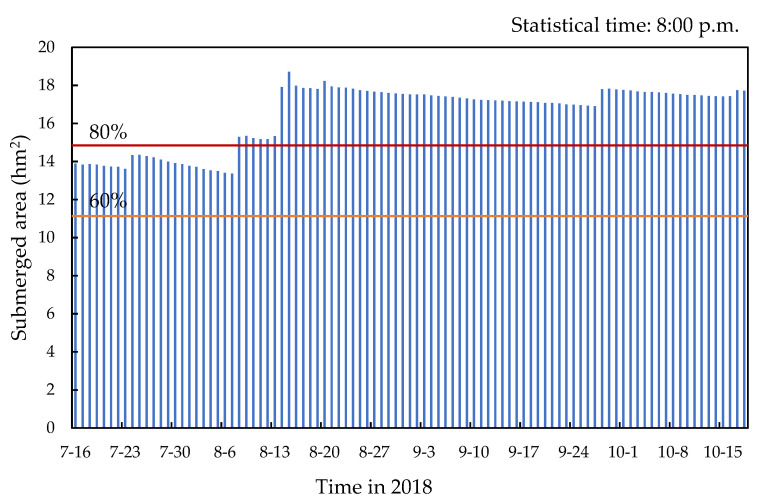
Histogram of inundation area of landscape water system in the study area during “the Expo Park”.

**Figure 14 ijerph-19-14349-f014:**
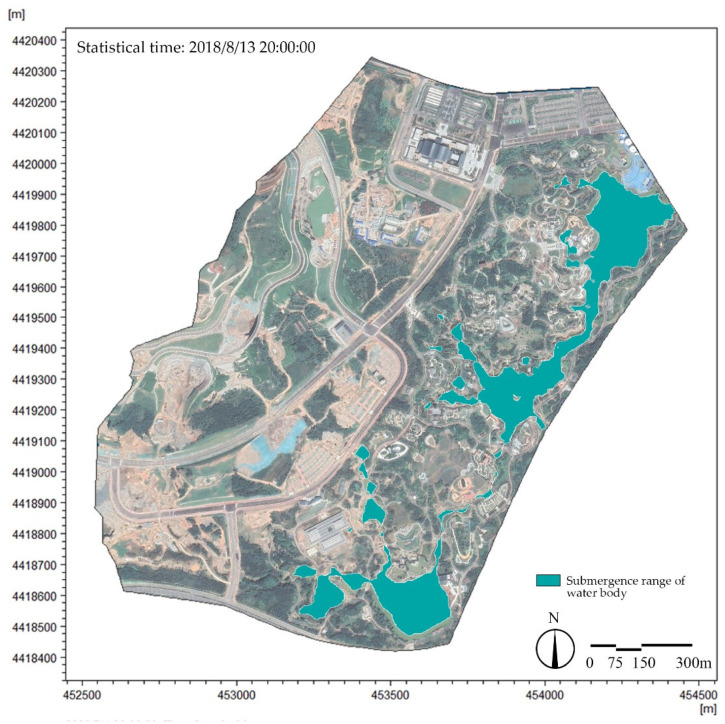
Distribution of submerged area of landscape water system in the study area. Note: The water system distribution map at other times is annexed.

**Table 1 ijerph-19-14349-t001:** Runoff coefficient of catchment area.

Underlying Surface Type		Area (hm^2^)	Rainfall Runoff Coefficient
External catchment	Mountain woodland	99.70	0.40
	Asphalt road	1.95	0.85
Internal catchment	Building roof	6.99	0.80
	Impervious paving	11.53	0.80
	Permeable pavement	8.74	0.30
	Green space	73.17	0.15
	Landscape water	19.73	1.00
Total		221.81	0.40

Rainfall runoff coefficient refers to the technical regulations for sponge city construction projects (DB13(J)/T 210-2016 [27]) and takes the value with the actual situation of the site.

**Table 2 ijerph-19-14349-t002:** Monitoring flow and simulated flow on 12 July 2021.

Time	Monitor Flow (m^3^/s)	Simulation Flow (m^3^/s)
8:00 p.m.	0.00	0.1
8:30 p.m.	0.00	0.15
9:00 p.m.	0.12	0.16
9:30 p.m.	0.38	0.48
10:00 p.m.	0.74	0.66
10:30 p.m.	1.01	1.15

**Table 3 ijerph-19-14349-t003:** Comparison of peak flow rate, peak reduction rate, and peak occurrence time of rainfall in different return periods before and after regional construction.

Rainfall		Peak Flow (m/s)	Time
2 years	Before	6.95	2 h 40 min
	After	0.00	None
	Difference (reduction rate)	6.95 (100.00%)	-
5 years	Before	11.45	2 h 45 min
	After	2.12	4 h 55 min
	Difference (reduction rate)	9.33 (81.48%)	2 h 5 min
10 years	Before	16.59	2 h 50 min
	After	4.52	4 h 15 min
	Difference (reduction rate)	12.07 (72.75%)	1 h 25 min
30 years	Before	26.26	2 h 50 min
	After	9.04	3 h 45 min
	Difference (reduction rate)	17.22 (65.58%)	55 min
50 years	Before	31.01	2 h 50 min
	After	11.01	3 h 40 min
	Difference (reduction rate)	20.00 (64.50%)	50 min
100 years	Before	37.63	2 h 50 min
	After	13.77	3 h 35 min
	Difference (reduction rate)	23.86 (63.41%)	45 min

Note: Rainfall is indicated with varying return periods; The time if the mean time of peak flow.

**Table 4 ijerph-19-14349-t004:** Comparison of outflow volume of rainfall with different return periods before and after construction of the sponge city.

Rainfall	Before (m^3^)	After (m^3^)	Reduction Rates (%)
2 years	32,640	0	100.00
5 years	53,750	10,180	81.06
10 years	73,340	25,980	64.58
30 years	108,640	58,510	46.14
50 years	126,090	74,710	40.75
100 years	151,140	99,930	33.88

Note: Rainfall is indicated with varying return periods; reduction rates refer to outflow volumes.

## Data Availability

The datasets generated during and/or analyzed during the current study are available from the corresponding author on reasonable request.

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
