# Peer review of "Effects of Low-Impact Development Facilities (Water Systems of the Park) on Stormwater Runoff in Shallow Mountainous Areas Based on Dual-Model (SWMM and MIKE21) Simulations"

_ijerph, 2022, doi:10.3390/ijerph192114349_

Round 1
Reviewer 1 Report
The research is interesting and the authors contribute with a solid methodological proposal. It is correctly written, with only two exceptional typographic mistakes in lines 15, 404 and 512.
The main contribution of the research is the methodological workflow in an interesting and current issue, as the problems due to artificialization and its consequences in soil sealing.
Then, I will explain below some suggestions to improve the manuscript with punctual changes.
1. First of all, the introduction is well-developed with many necessary concepts in the theoretical framework. Nevertheless, I suggest the authors should include a very close concept to their research and widely documented, as: soil sealing. Additionally, the authors mention the importance of growing of urban areas and sprawl, a concept commonly used to disperse and peri urban artificialization, meanwhile in this case the experience of sponge cities in China seems linked to large cities and urban cores. In this regard, a clarification in the introduction would be necessary. Finally, another key concept omitted in the theoretical framework is “ecosystem services” that authors may include in lines 60-79.
2. In keywords I suggest the authors to remove utility study which is excessive general, or even verification, in the same way.
3. Cartography improvement: In Figure 1, I suggest to delete “general layout” and include a locator map to locate the study area in the context of the city; in Figure 4 and Figure 5, the authors should adapt the legend symbols because in original format do not correspond to map symbols (size for instance) and legend background in white colour and outline to get a better reading. Figure 11 and Figure 14 should be reduced with the selection of some periods and include as annexe if necessary in present format. Additionally, in both figures text is too small and the quality or resolution low (pixelated).
4. Data collection. Here, authors should include more details about data, as format files or resolution (cell size) in raster layers.
5. Methodology. Figure 2 is interesting from the methodological point of view, but the text below (section 2.3.1…) seem structured under other criteria. Here I have a question to clarify methodology to readers: Is possible to design a methodological workflow aligned with the subsequent explanation?
6. Sections hierarchy. Authors should review section numeration because it is not homogeneous in present version. Example: section 2.3.1. with subsection “1” (in this case, can use a letter), section 3.1 with subsection “1” and by contrast section 3.2 with subsection “3.2.1”. A revision is necessary.
7. Position and numeration of figures. Some figures are disorder in the manuscript, or even cited below. A revision is necessary. For instance: Figure 9 should be cited after Table 3 according to the position in the manuscript, Figure 12 cited in line 459 corresponds to Figure 13.
8. Authors should improve discussion. In present form it is not at the necessary level of scientific paper. Many recent papers include the concept “sponge city”. Other experiences from other methods should be compared. Additionally, other reference case studies, as the district of Pudong should be cited too. In discussion or conclusion authors should clearly mention applicability or impact of their contribution, benefits in comparison with other approaches, replicability and exportability, among others.
Reviewer 2 Report
This finding of this research is interesting, but there are many parts needed to be improved. Comments for the authors are included below:
Abstract
1. Rewrite it concisely, the abstract should include a brief information about background, methods, results and concluding sentence.
Introduction
1- The introduction required significant rewriting. Please highlight the important information only not many general information.
2- The introduction fails to highlight the problem statement, or justify why need this modeling.
3- Highlight information about the application of SWMM for infiltration improvement such as in this recent papers: https://doi.org/10.2166/wcc.2021.180
4- The aim of the study is not clear, you mention it then cite other studies, which makes it not novel. Please state it clearly and concisely.
Materials and methods
1- Improve the quality of Figures, such as Figure 2.
2- How methods for calibration was conducted in the model?
3- How did you get the IDF curve for this study? Please cite the reference.
Results
1- The finding is interesting, but the results are lacking finding discussion with other studies. The discussion section seems lacking, I suggest you combine it with the results.
Conclusion
1- Please conclude the findings in according to the objectives.
2- Its long and need to be concise.
Author Response
Sincerely thank you for your valuable comments on the article, I have made improvements one by one. Details as follows:
Comments and Suggestions for Authors
Abstract
- Rewrite it concisely, the abstract should include a brief information about background, methods, results and concluding sentence.
Response 1:Thank you for your suggestion. The abstract (lines12-29) has been succinctly rewritten.
Introduction
1-The introduction required significant rewriting. Please highlight the important information only not many general information.
Response 1-:Thank you for your suggestion. The introduction has been rewritten to highlight key information and remove overly redundant sections (lines56-109).
2-The introduction fails to highlight the problem statement, or justify why need this modeling.
Response 2-:Thank you for your suggestions. The introduction has been reorganized to find the gaps in the existing research, which leads to the focus of the research in this paper (lines 56-109). In order to make the simulation results more comprehensive, so the dimensionality of the model is increased to combine the 1D model and 2D model to simulate the stormwater process at the site (lines 102-105).
3-Highlight information about the application of SWMM for infiltration improvement such as in this recent papers: https://doi.org/10.2166/wcc.2021.180
Response 3-:Thank you for your suggestion, the role of SWMM in infiltration improvement has been highlighted in the model construction section (lines 255-267).
4-The aim of the study is not clear, you mention it then cite other studies, which makes it not novel. Please state it clearly and concisely.
Response 4-:Thank you for your suggestion. The introduction has been reorganized to define the purpose of the study (lines 81-109).
Materials and methods
1-Improve the quality of Figures, such as Figure 2.
Response 1-:Thanks to your suggestions, the full text framework has been reorganized, a methodological workflow consistent with the follow-up has been designed, and the quality of the graphics has been improved (line 184).
2-How methods for calibration was conducted in the model?
Response 2-:Thank you for your question. The model calibration section has been reorganized and the process described in detail (lines 367-412).
3-How did you get the IDF curve for this study? Please cite the reference.
Response 3-:Thank you for your question. The IDF curves for this study were calculated using the Chicago method, and the full procedure has been added to the text (lines 235-246).
Results
1-The finding is interesting, but the results are lacking finding discussion with other studies. The discussion section seems lacking, I suggest you combine it with the results.
Response 1-:Thank you for your suggestions. In the results section, the article findings have been discussed in depth in relation to other studies (lines 584-629). Also, the necessary comparative discussion between shallow mountainous areas and urban areas has been added in the discussion section (lines 605-613).
Conclusion
1-Please conclude the findings in according to the objectives.
Response 1-:Thank you for your suggestion. The results have been described in a more focused manner (lines 631-656).
2-Its long and need to be concise.
Response 2-:Thank you for your suggestion. The presentation of the conclusion section has been streamlined (lines 631-656).
Thanks again.

Reviewer 3 Report
The manuscript presents a hydrologic and hydraulic study aiming to control the rainwater runoff in a shallow mountain area based on different synthetic storms. My main concern is that the manuscript reads more like a report than a scientific paper, missing essential elements of a scientific research. My major comments are:
1) The authors main goal as stated in the abstract (lines 18-19) is “to control the rainwater runoff in the study area, and solve the 18 problem of rainwater and flood in the periphery of the study area.” This is not really a scientific objective and reads more like an objective for the infrastructure itself or a technical report. Additionally, the introduction does not clearly identify the scientific objective of the study. I strongly encourage the authors to redesign their objective with a more rigorous scientific focus.
2) The introduction is not concise or objective, being unusual long for a scientific paper (1,500 words). The paragraph in lines 80-104 is unclear and must be improved to better justify the research gap. What type of quantitative analysis (lines 100-101) other studies don’t have that this manuscript proposes to do? There is no clear reference to this in the objective.
3) The introduction also contains elements that would be more appropriate to the methods section, such as lines 105-119 (model description) and 120-131 (study area). There is also duplicated information in the introduction and methods section about the study area and modeling softwares.
Data Collection section does not specify what monitoring data was collected in 12 July 2021 (line 184). Later in the text (line 341), the authors mention that they calibrated the model using such data. It is difficult to assess the validity of the methods with such imprecise description of the methods. Additionally, the model was calibrated based on a single event, without distinct events to validate the model, which is insufficient to support the suitability of the model to distinct events.
4) The authors refer to return period values multiple times as they can occur once every “X” years (lines 26, 301-303), which is a wrongly conception of the return period concept. For example, a value equivalent to a 2-year return period event is expected to be equaled or overcame at least once in 2 years. Meaning that that can occur more than once in this period.
5) There is basically no discussion (148 words) about the results of the paper and how they address the existing research gaps, which is a critical element of a scientific paper. The discussion (lines 476-484) only report limitations of the paper indicating that this research is incomplete.
In conclusion, addressing these major flaws in methodology requires overhaul of the paper and so I tend to recommend outright rejection.
Author Response
improvements one by one. Details as follows:
1) The authors main goal as stated in the abstract (lines 18-19) is “to control the rainwater runoff in the study area, and solve the 18 problem of rainwater and flood in the periphery of the study area.” This is not really a scientific objective and reads more like an objective for the infrastructure itself or a technical report. Additionally, the introduction does not clearly identify the scientific objective of the study. I strongly encourage the authors to redesign their objective with a more rigorous scientific focus.
Response 1:Thank you for your suggestions. The abstract and introduction have been rewritten (lines 12-109) and the study objectives have been redesigned (lines 16-18).
2) The introduction is not concise or objective, being unusual long for a scientific paper (1,500 words). The paragraph in lines 80-104 is unclear and must be improved to better justify the research gap. What type of quantitative analysis (lines 100-101) other studies don’t have that this manuscript proposes to do? There is no clear reference to this in the objective.
Response 2:Thank you for your suggestions. The introduction has been rewritten after sorting out the logical framework to concisely express the differences and innovations between this study and other studies (lines 56-109).
3) The introduction also contains elements that would be more appropriate to the methods section, such as lines 105-119 (model description) and 120-131 (study area). There is also duplicated information in the introduction and methods section about the study area and modeling softwares.
Response 3:Thank you for your suggestions. After combing through the entire text, the introduction section briefly mentions the stormwater management model (lines 99-100). A specific introduction to the model will be given in the modeling section (lines 248-253, 295-309).
Data Collection section does not specify what monitoring data was collected in 12 July 2021 (line 184). Later in the text (line 341), the authors mention that they calibrated the model using such data. It is difficult to assess the validity of the methods with such imprecise description of the methods. Additionally, the model was calibrated based on a single event, without distinct events to validate the model, which is insufficient to support the suitability of the model to distinct events.
Response :Thank you for your suggestion. Specific monitoring data have been added in the model calibration section of the text (lines 369-376). And the methodological steps are clearly described (lines 369-412). It is very unfortunate that due to various external conditions such as the epidemic, we were not able to visit the study area to conduct field data monitoring again. The absence of different events to validate the model is a major regret of this paper. This shortcoming has been stated in the discussion section of the paper and the reference to validation in the paper has been removed. This is an area that we need to improve in future studies.
4) The authors refer to return period values multiple times as they can occur once every “X” years (lines 26, 301-303), which is a wrongly conception of the return period concept. For example, a value equivalent to a 2-year return period event is expected to be equaled or overcame at least once in 2 years. Meaning that that can occur more than once in this period.
Response 4:Thank you for your suggestion. All parts of the text for the design reproduction period concept have been expressed correctly (lines 192-200, 239-243, 356-358, 443-447, 487).
5) There is basically no discussion (148 words) about the results of the paper and how they address the existing research gaps, which is a critical element of a scientific paper. The discussion (lines 476-484) only report limitations of the paper indicating that this research is incomplete.
Response 5:Thank you for your suggestion. Gaps between the findings of this paper and existing studies have been added in the Results section (lines 414-582). The contribution and applicability of this study has also been refined by adding a discussion section that highlights the differences between the results of the shallow mountainous and urban studies (lines 584-629).
Thanks again.

Round 2
Reviewer 1 Report
The authors addressed correctly all my suggestions. I consider the version of the manuscript is adequately improved.
Author Response
Sincerely thank you for your valuable comments on the article, which further improved the quality of the article.
Reviewer 3 Report
The authors addressed major issues in the previous version of the manuscript; however, the manuscript still reads more like a report than a scientific paper. These are my suggestions for improving the new version of the manuscript:
1) Objective:
The objective stated in the abstract seems to be different than the ones in the Introduction. The abstract should reflect the content of the manuscript. Additionally, the objectives still lack scientific rigor.
Line 79-82: Expand the model area is not a scientific objective. What do you want to achieve by expanding the model?
Line 82-83: What do you what to learn by comparing before and after construction scenarios?
Line 83-86: How these criteria will allow you to assess the LID effectiveness?
2) Introduction:
Aim for approximately 1,000 words.
Line 35-39: From my understanding, the focus of the paper is on flooding and not on aquifer recharging/water supply issues. Water security is a broad term that can have multiple meanings and used a few times along the manuscript. Define “water security” in the context of this study and focus on the problem that you aim to solve.
Line 43: reference missing for the policy implemented in China.
Line 43-56: This should be another paragraph
Line 58-59: Vague statement. What do you mean by “preliminary explorations”?
Line 70-75: LID should be properly introduced. What it is and what it is the importance for flood control. The literature review should focus on highlighting the role of LID on stormwater management and the lack of investigation of LID in developed mountain regions.
3) Discussion: The authors improved the discussion; however, it is still insufficient for a scientific paper. Please, see below.
Line 444: Water security needs to be better defined in the introduction. Vague statement. How does extreme precipitation can cause water security?
Line 444-453: This is a summary of the paper, not discussion. It is ok to have some summary here; however, it should help the reader to get to the key points that are discussed. There is no explanation about this research gap on single-model simulations. Is this actually a key point from this study?
Line 456-457: Be specific about the benefits of having such low velocities.
Line 464: What differences? Be specific.
Line 467-467: Why great importance? Be specific.
Line 467-469: The role of LID must be well described in the introduction.
Line 464-472: This paragraph does not contain relevant discussion and should be rewritten. What are the key points from this study that should be highlighted and how it compares with the literature.
Line 473: If this paper failed, why publish it? Rewrite.
Author Response
Sincerely thank you for your valuable comments on the article, which further improved the quality of the article. I have made improvements one by one, as detailed below:
The authors addressed major issues in the previous version of the manuscript; however, the manuscript still reads more like a report than a scientific paper. These are my suggestions for improving the new version of the manuscript:
1) Objective:
The objective stated in the abstract seems to be different than the ones in the Introduction. The abstract should reflect the content of the manuscript. Additionally, the objectives still lack scientific rigor.
Thank you for your suggestion. The objectives have been improved and checked for consistency between the objectives in the abstract and the introduction (lines16-19, lines122-126).
Line 79-82: Expand the model area is not a scientific objective. What do you want to achieve by expanding the model?
Thank you for your suggestion. The objectives have been improved (lines 122-126). We would like to extend the model to study the role of LID facilities on flooding while using stormwater reuse to enhance the water landscape (lines 122-126 and lines 493-499).
Line 82-83: What do you what to learn by comparing before and after construction scenarios?
Thank you for your question. We investigate the effect of LID facilities on flood reduction and stormwater reuse under different rainfall conditions by comparing pre-and post-construction scenarios (lines 122-126).
Line 83-86: How these criteria will allow you to assess the LID effectiveness?
Thank you for your question. These criteria were introduced to assess the performance of LID facilities at two levels: flood risk and stormwater reuse to replenish landscape water systems (lines 129-131). In the results section, the effectiveness of LID is assessed by comparing each of the five criteria post-construction to pre-construction in shallow mountainous areas, whether their post-construction results reduce flood risk, and whether they fully utilize stormwater reuse to supplement landscape water systems (lines 370-489).
2) Introduction:
Aim for approximately 1,000 words.
Thank you for your suggestion. The content of the introduction has been enriched, and the final introduction section is 951 words (lines 36-132).
Line 35-39: From my understanding, the focus of the paper is on flooding and not on aquifer recharging/water supply issues. Water security is a broad term that can have multiple meanings and used a few times along the manuscript. Define “water security” in the context of this study and focus on the problem that you aim to solve.
Thank you for your suggestion. In the context of this study, water security means that surface runoff from shallow mountainous areas under various rainfall conditions does not pose a threat to shallow mountainous areas and cities. The water security issue is flood risk. The reference to water security issues in the text has been changed to flood risk (lines 174, 181, 596).
Line 43: reference missing for the policy implemented in China.
Thank you for your suggestion. Official circulars of policies implemented in China have been added as well as references to policy implementation standards (lines 58-71). Among them: "Notice on Systematic and Territory-wide Demonstration of Sponge City Construction" indicates the future direction of sponge city development in China. The Technical Guide for Sponge City Construction - Construction of Low-Impact Development Stormwater Systems (for Trial Implementation) is the construction standard.
Line 43-56: This should be another paragraph
Thank you for your suggestion. A separate paragraph has been created (lines 51-71).
Line 58-59: Vague statement. What do you mean by “preliminary explorations”?
Thank you for your suggestion. A more accurate representation has been used (lines 108-110).
Line 70-75: LID should be properly introduced. What it is and what it is the importance for flood control. The literature review should focus on highlighting the role of LID on stormwater management and the lack of investigation of LID in developed mountain regions.
Thank you for your suggestion. The concept and principles of LID facilities were added to the introduction section. And the literature review was rewritten to emphasize the role in stormwater management as well as to cite the lack of LID studies in developed mountain areas (lines 78-121).
3) Discussion: The authors improved the discussion; however, it is still insufficient for a scientific paper. Please, see below.
Line 444: Water security needs to be better defined in the introduction. Vague statement. How does extreme precipitation can cause water security?
Thank you for your suggestion. The reference to water security issues in the text has been changed to flood risk (lines 174, 181, 596). The causes of flood risk due to extreme precipitation have been added to the discussion section (lines 491-493).
Line 444-453: This is a summary of the paper, not discussion. It is ok to have some summary here; however, it should help the reader to get to the key points that are discussed. There is no explanation about this research gap on single-model simulations. Is this actually a key point from this study?
Thank you for your suggestion. The discussion section has been rewritten (lines 490-499).
Line 456-457: Be specific about the benefits of having such low velocities.
Thank you for your suggestion. The benefits of low speed have been added to the discussion section (lines 520-524).
Line 464: What differences? Be specific.
Thank you for your suggestion. The difference in performance between shallow mountain and urban LID facilities has been detailed in the discussion section (lines 500-529).
Line 467-467: Why great importance? Be specific.
Thank you for your suggestion. Already added in the discussion section (lines 520-529).
Line 467-469: The role of LID must be well described in the introduction.
Thank you for your suggestion. A description related to the LID facility has been added in the introduction section (lines 78-94).
Line 464-472: This paragraph does not contain relevant discussion and should be rewritten. What are the key points from this study that should be highlighted and how it compares with the literature.
Thank you for your suggestion. The discussion section has been rewritten. The discussion focuses on the performance of shallow mountain LID facilities through comparison with other literature (lines 500-529).
Line 473: If this paper failed, why publish it? Rewrite.
Thank you for your suggestion. The possibility of failure of this study has already been addressed in the Discussion section. The implications for the study after failure are also explored (lines 568-572).
